# Experimental Study on the Adhesion of Abalone to Surfaces with Different Morphologies

**DOI:** 10.3390/biomimetics9040206

**Published:** 2024-03-29

**Authors:** Peng Xi, Yanqi Qiao, Qian Cong, Qingliang Cui

**Affiliations:** 1College of Agricultural Engineering, Shanxi Agricultural University, Jinzhong 030801, China; yydljxp2000@126.com (P.X.); 20233053@stu.sxau.edu.cn (Y.Q.); 2Dryland Farm Machinery Key Technology and Equipment Key Laboratory of Shanxi Province, Shanxi Agricultural University, Jinzhong 030801, China; 3College of Biological and Agricultural Engineering, Jilin University, Changchun 130022, China; congqian@jlu.edu.cn; 4Key Laboratory of Bionic Engineering Ministry of Education, Jilin University, Changchun 130022, China

**Keywords:** abalone, abdominal foot, adhesion, non-smooth surface, force measuring plate

## Abstract

To date, research on abalone adhesion has primarily analyzed the organism’s adhesion to smooth surfaces, with few studies on adhesion to non-smooth surfaces. The present study examined the surface morphology of the abalone’s abdominal foot, followed by measuring the adhesive force of the abalone on a smooth force measuring plate and five force measuring plates with different surface morphologies. Next, the adhesion mechanism of the abdominal foot was analyzed. The findings indicated that the abdominal foot of the abalone features numerous stripe-shaped folds on its surface. The adhesion of the abalone to a fine frosted glass plate, a coarse frosted glass plate, and a quadrangular conical glass plate was not significantly different from that on a smooth glass plate. However, the organism’s adhesion to a small lattice pit glass plate and block pattern glass plate was significantly different. The abalone could effectively adhere to the surface of the block pattern glass plate using the elasticity of its abdominal foot during adhesion but experienced difficulty in completely adhering to the surface of the quadrangular conical glass plate. The abdominal foot used its elasticity to form an independent sucker system with each small lattice pit, significantly improving adhesion to the small lattice pit glass plate. The elasticity of the abalone’s abdominal foot created difficulty in handling slight morphological size changes in roughness, resulting in no significant differences in its adhesion to the smooth glass plate.

## 1. Introduction

Over a long period of evolution, organisms have developed unique and exceptional adaptations to thrive in their natural environments. For example, many animals in nature possess adhesive capabilities [1,2,3]. Animals are able to firmly adhere to different surfaces in their environment using their adhesive abilities, helping such organisms with fundamental survival tasks like crawling, hunting, grabbing, and fleeing [4,5,6,7,8]. Adhesion is not only used by many animal varieties in nature but also plays an important role in human production and life. The most typical applications of this organism involve vacuum suckers, which are widely used in industrial production and people’s daily lives through the adhesion of different pressures inside and outside the sucker [9,10,11,12]. However, vacuum suckers have high requirements for the adhesion surface and offer good adhesion only on smooth surfaces. The adhesion effect on non-smooth surfaces is poor or absent. At the same time, this adhesion effect is prone to leakage failure, resulting in accidents [13,14,15]. However, organisms with adhesion capabilities not only adhere to smooth surfaces but also yield good adhesion effects on non-smooth surfaces. The ability to produce strong adhesion effects on both smooth and non-smooth surfaces has attracted great interest from relevant researchers. These researchers have selected organisms with adhesion capabilities and observed the structures of their biological suckers in detail. The adhesion capabilities of biological suckers onto different morphological surfaces have also been measured using experimental methods. This study seeks to improve the poor adhesion capabilities of vacuum suckers onto non-smooth surfaces using the method of engineering bionics, thus facilitating the development of vacuum suckers.

To date, researchers have studied and achieved results related to the adhesion capabilities of some common adhesive organisms such as the octopus, leech, remora, tree frog, and Northern clingfish. Tramacere et al. studied octopus suckers and found that each sucker was divided into upper and lower chambers, with the two chambers connected through an orifice in the center. When the sucker is adsorbed, the lower chamber first adheres to the surface and then gradually flattens to increase the adhesion area. Finally, the sucker forms a sealing structure with the surface. The upper cavity of the sucker has a protruding structure covered by a large number of fibers. The water in the lower chambers of the sucker is extruded through the orifice into the upper cavity via the gradual extrusion of the lower chambers of the sucker. At the same time, the protruding structure of the upper cavity and the orifice form an effective seal, ultimately producing vacuum pressure that enables the sucker to adhere to the object’s surface. The fiber structure on the surface of the protrusion can improve the sealing ability between the protrusion and the orifice [16,17,18]. Ditsche et al. observed the abdominal suckers of Northern clingfish, which have adhesion capabilities, and found that the suckers’ surfaces were composed of numerous micron-sized fibers with different size grades. When Northern clingfish adhere to a surface using their suckers, these fibers ensure that a sealed structure is produced regardless of the roughness of the surface, enabling the adhesion of Northern clingfish to different surfaces in nature. Adhesion experiments showed that the Northern clingfish has good adhesion (30–40 kPa) and adaptability to non-smooth surfaces [19,20]. Chuang et al. observed the abdominal sucker of the Pulin river loach (*Sinogastromyzon puliensis*) with adhesion capabilities and found that its surface was composed of many radial fins. The surfaces of the radial fins also had micron-sized fiber structures. When the Pulin river loach adhere to a surface, the radial fins and fiber structure of the abdominal sucker enable the adhesion of the organism to non-smooth surfaces [21]. Kampowski et al. observed the suckers of leeches with adhesion capabilities with scanning electron microscopy and found that a large number of pores were distributed on the surfaces of the suckers in front of and behind the leech. When the leech engages in adsorption, the small holes on the sucker can secrete mucus to fill the unevenness of the adhesion surface, improving the sealing performance of the leech sucker when the surface roughness is larger. This mechanism increases the adhesion force of the sucker. Abalone is an adhesive organism in the ocean whose abdominal foot has strong adhesion capabilities [22,23,24]. According to reports, an abalone with a body length of about 15 cm has an adhesion force of up to 200 kg, highlighting the organism’s strong adhesion force [25]. Due to the strong adhesive properties of abalone, researchers have conducted extensive studies on the adhesion of the organism’s muscular foot. Lin et al. studied the American red abalone and found that its abdominal foot surface is composed of fibers with two sizes. This multi-level fiber structure enables the abdominal foot sucker to form an interlocking structure on surfaces with a variety of roughness types, effectively improving the adaptability of abalone to different adhesion surfaces [26]. Li et al. tested the adhesion force of abalone in both water and air using various force measuring plates. The authors found that the adhesion force of the abalone’s abdominal foot primarily comes from vacuum adhesion force, van der Waals force, and capillary force [27]. Xi analyzed the measurement results of abalone’s adhesion force on different force measuring plates and determined that the vacuum adhesion force plays a significant role in the total adhesion force of abalone [28].

To date, research on abalone has mainly focused on the adhesion capabilities of the organism’s abdominal foot on smooth surfaces, as well as the composition of the adhesion force and the surface structure of abalone’s abdominal foot. However, there are few studies on the adhesion capabilities of abalone’s abdominal foot on non-smooth surfaces. To fill this gap, the present study offers a new direction for the bionic design of vacuum suckers by studying the abalone’s adhesion capabilities and modes of action on non-smooth surfaces. The specific research contents are as follows. Firstly, abalone samples of basically the same mass and size were selected for feeding, and then the surface morphology of the abalone’s abdominal foot was observed macroscopically and microscopically. Force measuring plates with different surface morphologies were then selected, and the adhesion force of the abalone’s abdominal foot on force measuring plates with different surface morphologies was measured via tensile testing. The corresponding adhesion stress was obtained according to the area of the abalone’s abdominal foot. The effects of force measuring plates with different surface morphologies on the adhesion of the abalone’s abdominal foot were compared, and the mode of action between them was explored. This paper provides a reference for studying the adhesion capabilities of other organisms with adhesion capabilities on non-smooth surfaces and the interactions between them.

## 2. Materials and Methods

### 2.1. Observation of the Abalone’s Abdominal Foot

#### 2.1.1. Abalone Sample Preparation

The abalone used in this experiment was *Haliotis discus hannai*, which was acquired from an aquatic market and promptly transferred to a laboratory aquarium for feeding. The aquarium measured 1500 × 1000 × 600 mm and was equipped with a filtration system and water circulation system. The water temperature in the tank was maintained between 15 and 20 °C, with a salinity of 30% and a water depth of 0.5 m. The abalone samples were nourished with wakame to ensure their survival in the aquarium [29,30]. The abalone samples weighed between 50 and 65 g and were acclimated in the aquarium for a minimum of 10 days before the experiment to mitigate errors stemming from individual variations.

#### 2.1.2. Observations of the Abalone’s Abdominal Foot Surface Morphology

The main structure of abalone is shown in Figure 1a,b and is composed of a hard shell with soft abdominal feet. Figure 1c presents the positional relationship between the abdominal foot and the shell. The abdominal foot serves as the primary organ responsible for abalone’s adhesion and crawling. To facilitate further research on abalone adhesion, the surface morphology of the abdominal foot was examined using a stereomicroscope (Stemi 2000-C, ZEISS, Oberkochen, Germany). The surface morphology of an abalone’s abdominal foot is shown in Figure 1d, with the abdominal foot surface segmented into three layers: the outer layer, the middle layer, and the inner layer. The inner layer, which encompasses the majority of the abdominal foot area, displays numerous striped folds on its surface, as shown in Figure 1e. The abdominal foot has a certain degree of elasticity and stretch through the different areas of the abdominal foot’s striped folds, generated by driving forward movement.

### 2.2. Adhesion Test

#### 2.2.1. Preparation of the Force Measuring Plate

Using a tensile test, we measured the adhesion of the abalone’s abdominal foot to force measuring plates with different surface morphologies. Due to the good adhesion and adaptability of the abalone’s abdominal foot to the glass plate, this surface enabled us to observe changes in the abalone’s abdominal foot. Thus, the glass plate was selected as the force measuring plate for the tensile test. In this paper, six types of glass plates with different surface morphologies were selected: (1) a smooth glass plate; (2) a fine frosted glass plate, roughness Ra = 0.86 μm; (3) a coarse frosted glass plate, roughness Ra = 480 μm; (4) a quadrangular conical glass plate with a side length of 1.5~5 mm and a height of 1 mm, as shown in Figure 2b; (5) a block pattern glass plate with a block side length of 10~20 mm and height of 0.5 mm, as shown in Figure 2c; and (6) a small lattice pit glass plate with a pit length of 0.8 mm, as shown in Figure 2d. The surface morphologies of the six force measuring plates are shown in Figure 2a.

#### 2.2.2. Design and Processing of the Hook

To measure the adhesion force of the abalone, we had to detach the abalone in its adhesion state from the force measuring plate, as the adhesion force of the abalone’s abdominal foot is notably strong. Moreover, abalone shell shapes present certain differences between individuals. For this purpose, we designed a type of hook that could hook the shell without affecting the adhesion of the abalone. The 3D design model of the hook and relevant design details are shown in Figure 3A, as follows: (a) a hole diameter of 5 mm; (b) a concave design to avoid contact with the abalone shell; (c) a certain radian designed to adapt to the abalone shell; (d) a shallow groove design conveniently able to catch the abalone shell; (e) a chamber structure designed to prevent scratching the abalone; and (f) a wedge structure designed for easy insertion into the abalone shell while increasing strength. Then, the hook was processed via 3D printing (UP! three-dimensional printer) with PLA used as the printing material. The 3D-printing process of the hook is shown in Figure 3B, and the solid machined hook is shown in Figure 3C.

#### 2.2.3. Abalone Abdominal Foot Adhesion Test

The universal testing machine (WSM-500N) used in the tensile test was controlled with a computer. Before the test, six force measuring plates were placed at the bottom of each leaching basin. Then, one abalone (purchased and kept in the aquarium, as shown in Section 2.1.1) was placed on each force measuring plate. Finally, the leaching basin was placed at the bottom of the aquarium together with the abalone samples on the force measuring plates, which were left to slowly adhere. In each test, the force measuring plate and abalone adhering to its surface were placed together on the testing machine for the tensile test. In this test, the force measuring plate was fixed first, and then the left and right sides of the abalone shell were hooked with two self-made hooks. Next, the tensile test was carried out. Figure 4 presents a schematic diagram of the tensile test. In this test, the lifting speed of the tensile testing machine was 100 mm/min. As the abalone became subjected to increasing upward tension, the abdominal foot was gradually separated from the force measuring plate. The test ended when the abalone’s abdominal foot was completely detached from the force measuring plate. Then, the maximum tensile force on the abalone during the whole tensile process was recorded and taken as the adhesion force of the abalone’s abdominal foot. The time interval between each tensile test was 24 h to ensure that the adhesion force of the abdominal foot did not increase with extended time. Five tests were performed for each force measuring plate.

## 3. Results and Discussion

### 3.1. Calculation and Analysis of Test Results

Table 1 presents the adhesion force of the abalone on the six force measuring plates along with the corresponding mass values. To analyze the ability of the abalone’s abdominal foot to adhere to a surface, the relevant adhesion stress (f) must be determined. The adhesion stress (f) is defined as
f = F/A(1)
where F represents the adhesion force of the abalone on the force measuring plate, and A is the corresponding abalone’s abdominal foot area. Since an abalone is a living creature, it was not possible to artificially unfold the abalone’s abdominal foot and force it to adhere onto the force measuring plate. Indeed, each abalone’s abdominal foot was generally curled up, as shown in Figure 1b, making it difficult to measure the area of the abdominal foot when adhered onto the force measuring plate immediately after each tensile test. Due to the challenges in directly measuring the abdominal foot area during adhesion in this experiment, we instead measured the mass of the abalone and calculated the corresponding abdominal foot area. Ten abalones used in the experiment were randomly selected. The mass (g) of each abalone and the corresponding area (mm^2^) of its abdominal foot were measured separately. The ratio (S) of the abdominal foot area (mm^2^) to mass (g) was calculated separately for each abalone based on measurements, and then the average of ratio (S) for 10 abalones was calculated as S_AVG_. The average S_AVG_ value was 43.15. Thus, the abdominal foot area (A) of the abalone was calculated in this experiment by measuring the abalone’s mass and multiplying the ratio, S_AVG_.

Table 1 presents the tensile test results and corresponding abalone mass. The calculated corresponding adhesion stress value (f) of the abalone’s abdominal foot on the six different force measuring plates is shown in Table 2.

Figure 5 presents a box plot of the average adhesion stress (f) of the abalone’s abdominal foot on different force measuring plates. It can be seen from Figure 5 that the average values of adhesion stress (f) on the smooth glass plate, fine frosted glass plate, coarse frosted glass plate, and quadrangular conical glass plate were basically the same, whereas the stress value on the block pattern glass plate was slightly larger. The stress value on the small lattice pit glass plate was largest.

### 3.2. Adhesion Mechanism Analysis

The adhesion stresses of the abalone on five force measuring plates with different roughness types and surface morphologies (Table 2) were analyzed for significance (a significance level of 0.05) to comparatively analyze the effects of the plates on the adhesion of the abalone’s abdominal foot. A rank sum test was used to compare the values of five abalone adhesion stresses on a fine frosted glass plate, a coarse frosted glass plate, a small lattice pit glass plate, a quadrangular conical glass plate, and a block pattern glass plate with those on a smooth glass plate, respectively. The results are shown in Table 3.

The significance analysis results in Table 3 show that the adhesion stress (f) of abalone on the fine frosted glass plate, coarse frosted glass plate, and quadrangular conical glass plate was not significantly different from that on the smooth glass plate (*p* > 0.05). The adhesion stress on the glass plate with small lattice pits and that on the block pattern glass plate were significantly different, indicating that the adhesion stress (f) of the abdominal foot on these two force measuring plates was significantly different from that on the smooth glass plate (*p* < 0.05).

The adhesion stress of abalone on the block pattern glass plate was significantly different from that on the smooth glass plate, primarily because the surface morphology of the block pattern glass plate changed slowly, with blunt corners. This slow rate of change better enabled the abalone’s abdominal foot to exert its stretching capabilities. The abalone abdominal foot can completely adhere to the morphology surface of the block pattern force measuring plate, as shown in Figure 6a. The adhesion area of the abalone was larger than the area of the abdominal foot upon complete attachment to the morphological surface of the block pattern glass plate, thereby increasing the adhesion force and the adhesion stress of the abalone on its surface, as shown in Figure 6b. The shape of the quadrangular conical glass plate did not have a significant impact on the adhesion of the abalone’s abdominal foot, primarily because the quadrangular cone itself in the quadrilateral conical glass plate and the shape between the quadrangular cone changed rapidly; i.e., the rotation angle was sharp, and the ridges were too numerous. As a result, it was difficult for the abalone’s abdominal foot to exert its stretching characteristics and completely adhere to the morphological surface of the quadrilateral conical glass plate, as shown in Figure 7a. As can be seen from Figure 7a, there are not many clear quadrangular conical imprints on the surface of the abalone abdominal foot, indicating that the elasticity and stretching capabilities of the abalone abdominal foot does not allow it to adhere well to the surface of the quadrangular conical glass plate. The adhesion area of the abalone to the force measuring plate was essentially the same as the size of the organism’s abdominal foot area. Thus, the adhesion stress of the abalone on the plate’s surface did not change significantly, as shown in Figure 7b. 

The adhesion stress of abalone on the small lattice pit glass plate was significantly higher than that on the smooth glass plate because, under normal circumstances, the abalone formed a sucker structure on the smooth force measuring plate, as shown in Figure 8a. Because an abalone is a living creature, according to the experiment, the degree of the vacuum between the abdominal foot and the force measuring plate was far less than 100% (about 40%) when adhering to the smooth force measuring plate. When the abalone adhered to the force measuring plate with small, shallow lattice pits, the abdominal foot, which has certain elasticity, wrapped around the small lattice pit and excluded some of the gas in the pit. Figure 8b shows the adhesion state of the abalone abdominal foot on the small lattice pit glass plate, and it can be seen that there are a lot of small squares on the surface of the abdominal foot, which is formed by the abalone abdominal foot due to the elastic deformation squeezed into the small pits. In this way, each small lattice pit formed a separate sucker structure, as shown in Figure 8c. As a result, the vacuum degree of the whole abdominal foot on the force measuring plate with small lattice pits was significantly increased, so the adhesion stress of the abalone on the force measuring plate with small lattice pits was greater than that on the smooth force measuring plate.

The results on the various measuring plates (Table 2) along with the significance analysis results (Table 3) showed that the adhesion stress of the abalone’s abdominal foot did not change significantly with an increase in the roughness of the force measuring plate (i.e., the smooth glass plate, fine frosted glass plate (Ra = 0.86 μm), and coarse frosted glass plate (Ra = 480 μm)). This result shows that the elasticity and stretch of the abdominal foot sucker create difficulties in adhering to small surface morphologies of roughness dimension levels (micron-sized). The adhesion area of the abalone on the force measuring plate was essentially the same as the size of the abdominal foot. Thus, the increase in the roughness of the force measuring plate did not yield an increase in the adhesion of the abdominal foot.

In this paper, the mechanism analysis of abalone adhesion only considered the effect between the elasticity of the abdominal foot and the surface morphology of different force measuring plates, and other factors were not considered. Therefore, the analysis of the adhesion mechanism of abalone on different force measuring plates is only a hypothesis, which needs further experiments to confirm.

## 4. Conclusions

Through observations and tensile tests, we studied the effects of using force measuring plates with different surface morphologies on the adhesion of the abalone’s abdominal foot, and corresponding conclusions were drawn.

(1)There was no significant difference in the adhesion of the abalone to the fine frosted glass plate, coarse frosted glass plate, quadrangular conical glass plate, or smooth glass plate. However, adhesion to the small lattice pit glass plate and block pattern glass plate was significantly different.(2)The quadrangular conical shape in the quadrangular conical glass plate changed rapidly, making it difficult for the abalone’s abdominal foot to fully adhere to the morphological surface of this plate. Conversely, the surface morphology of the block pattern glass plate changed slowly, enabling the abalone’s abdominal foot to fully adhere to this plate’s surface. When the abalone adhered to the small lattice pit glass plate, each small lattice pit was enclosed, excluded some of the gas in the pit, forming an independent sucker system due to the stretching characteristics of the abdominal foot and resulting in a significant increase in the adhesion of the abdominal foot.(3)Changes in the stretching of the abdominal foot created difficulties in achieving small morphological size changes based on the roughness, leading to no significant differences in the adhesion of abalone to force measuring plates with different types of roughness.

## Figures and Tables

**Figure 1 biomimetics-09-00206-f001:**
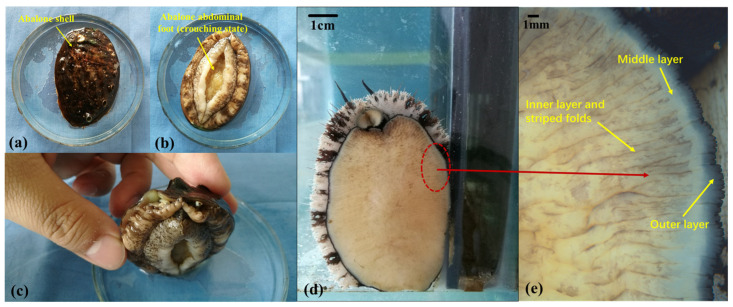
(**a**) Abalone shell; (**b**) abalone abdominal foot (crouching state); (**c**) the positional relationship between the abdominal foot and the shell; (**d**) abalone abdominal foot surface; (**e**) three layers and striped folds of the abalone’s abdominal foot.

**Figure 2 biomimetics-09-00206-f002:**
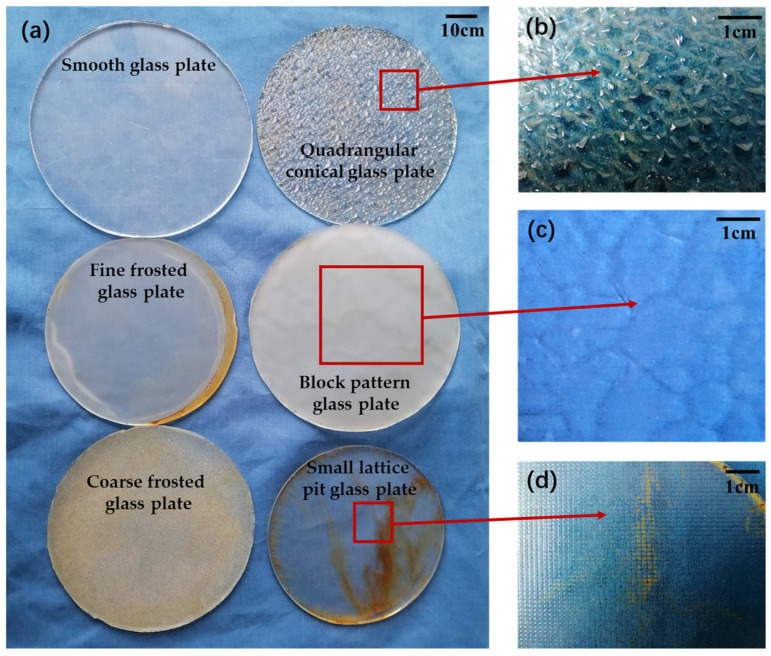
(**a**) The six force measuring plates used for the tensile test; (**b**) the quadrangular conical glass plate and the specific quadrangular conical morphology; (**c**) the block pattern glass plate and block pattern’s specific morphology; (**d**) the small lattice pit glass plate and the pit’s specific morphology.

**Figure 3 biomimetics-09-00206-f003:**
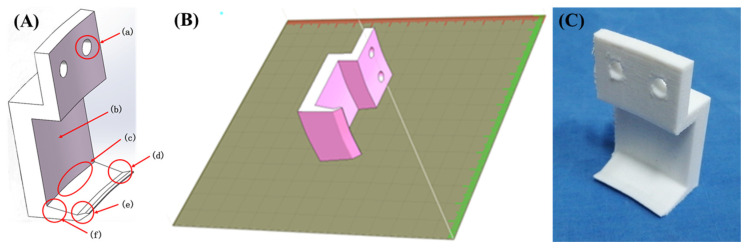
(**A**) Hook three-dimensional model and design details; (**B**) 3D-printing process of the hook; (**C**) 3D-printing hook entity.

**Figure 4 biomimetics-09-00206-f004:**
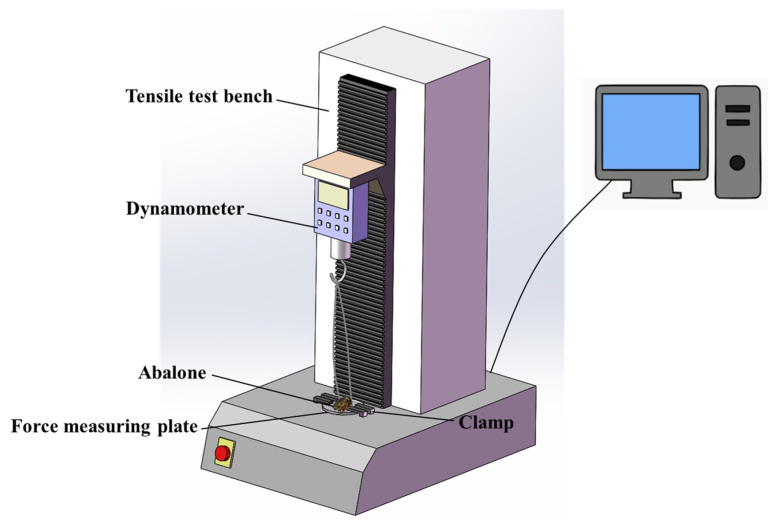
Tensile test schematic diagram.

**Figure 5 biomimetics-09-00206-f005:**
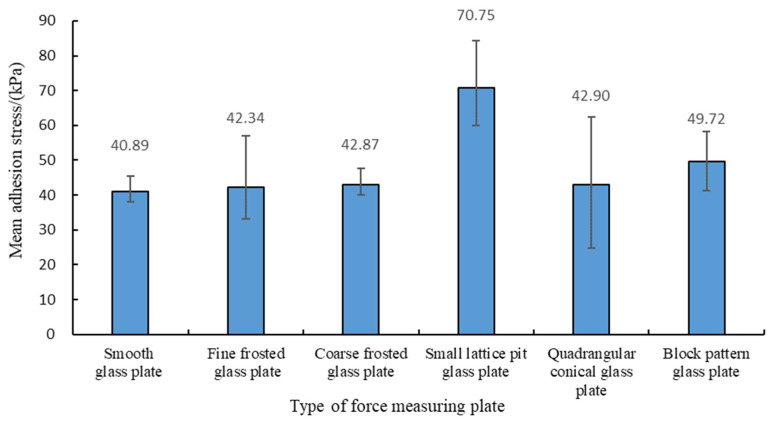
The average adhesion stress of the abalone’s abdominal foot on the six different force measuring plates.

**Figure 6 biomimetics-09-00206-f006:**
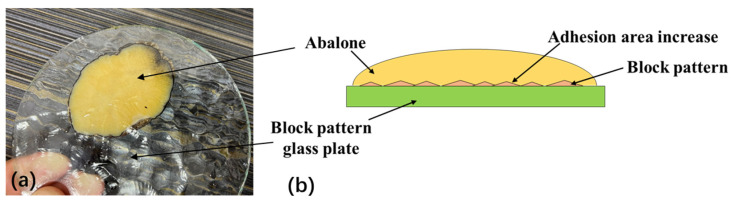
(**a**) The adhesion of the abalone’s abdominal foot to the block pattern glass plate; (**b**) The adhesion area of the abalone was larger than the area of the abdominal foot upon complete attachment to the morphological surface of the block pattern glass plate.

**Figure 7 biomimetics-09-00206-f007:**
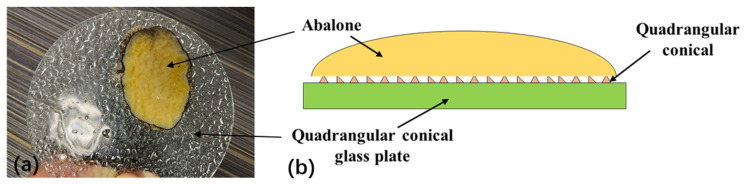
(**a**) The adhesion of the abalone’s abdominal foot to the quadrangular conical glass plate; (**b**) The adhesion area of the abalone to the quadrangular conical glass plate was essentially the same as the size of the abdominal foot area.

**Figure 8 biomimetics-09-00206-f008:**
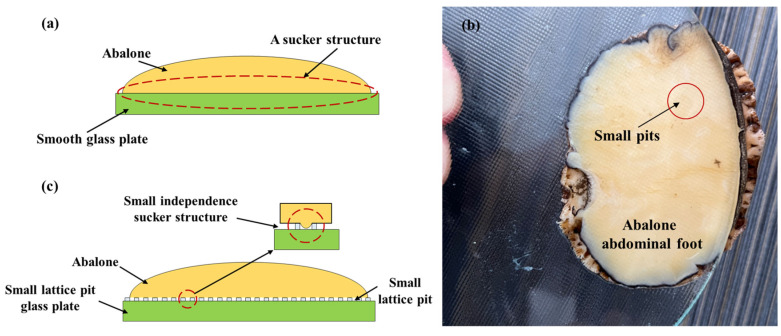
(**a**) The abalone formed a sucker structure on the smooth force measuring plate; (**b**) The adhesion of the abalone’s abdominal foot to the small lattice pit glass plate; (**c**) Each small lattice pit of the small lattice pit glass plate formed a independence sucker structure with the abalone’s abdominal foot.

**Table 1 biomimetics-09-00206-t001:** Adhesion force and corresponding mass of the abalone on the six force measuring plates.

Maximum Adhesion Force/N	Type of Force Measuring Plate					
Test Times	Smooth Glass Plate	Fine Frosted Glass Plate	Coarse Frosted Glass Plate	Small Lattice Pit Glass Plate	Quadrangular Conical Glass Plate	Block Pattern Glass Plate
1	80.5	110.9	105.2	176.1	61.17	126.6
Abalone mass/g	49.1	67.5	54.9	56	57.4	60.2
2	89.13	95.92	89.46	149.5	175.6	125.1
Abalone mass/g	48.3	67	51.5	57.2	65.3	60.3
3	116.9	142.9	103.2	204	108.2	113.3
Abalone mass/g	59.7	58.1	56.9	56	56	63.6
4	96.9	114.6	112.7	144.9	94.7	129.4
Abalone mass/g	57.4	60.3	54.7	56	59.4	51.5
5	101.6	102.5	92.64	191.6	114.5	116.1
Abalone mass/g	60.1	60.3	53.8	58.5	58	51.5

**Table 2 biomimetics-09-00206-t002:** Adhesion stress (f) of abalone on the six force measuring plates.

Adhesion Stress/kPa	Type of Force Measuring Plate					
Test Times	Smooth Glass Plate	Fine Frosted Glass Plate	Coarse Frosted Glass Plate	Small Lattice Pit Glass Plate	Quadrangular Conical Glass Plate	Block Pattern Glass Plate
1	38.00	38.08	44.41	72.88	24.70	48.74
2	42.77	33.18	40.26	60.57	62.32	48.08
3	45.38	57.00	42.03	84.42	44.78	41.28
4	39.12	44.04	47.75	59.97	36.95	58.23
5	39.18	39.39	39.91	75.90	45.75	52.24
Average value	40.89	42.34	42.87	70.75	42.90	49.72

**Table 3 biomimetics-09-00206-t003:** The significance analysis results of the abalone’s adhesion stress on the six different force measuring plates.

Type of Force Measuring Plate	*p* Value	Explanation
Fine frosted glass plate	0.917	Comparison with smooth glass plate
Coarse frosted glass plate	0.251	Comparison with smooth glass plate
Small lattice pit glass plate	0.009	Comparison with smooth glass plate
Quadrangular conical glass plate	0.754	Comparison with smooth glass plate
Block pattern glass plate	0.028	Comparison with smooth glass plate

## Data Availability

Data are contained within the article.

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
