# Peer review of "Experimental Study on the Adhesion of Abalone to Surfaces with Different Morphologies"

_biomimetics, 2024, doi:10.3390/biomimetics9040206_

Round 1

Reviewer 1 Report

Comments and Suggestions for Authors

In this manuscript, the authors present a study of the adhesive performance of abalone on various substrates with different roughnesses. While the study can help contribute to our understanding of how abalone adhesion is generated, there are many crucial details missing in the presentation of the results and additional experiments are required to confirm the hypotheses on the mechanisms of adhesion presented by the authors (Figures 10-12). Please see below for my detailed feedback, starting with major issues then minor issues.

Major issues:

- The writing needs to be greatly improved. There are a lot of vague statements that don't contribute anything useful to the manuscript. Below, in the 'minor issues', I point out some examples, but there are many more throughout the text.

- I would find it useful if the authors provide an image or schematic of the organism and its adhesive foot, which could complement the Introduction. Figure 1 provides two images, but there is no perspective regarding where this foot is on the organism, relative to its general bodyplan and morphology.

- For clarity and conciseness, Figures 2, 3, 4, and 5 could be combined into one single figure. Each picture should also include a scale bar.

- Figures 6-7 should be combined as well. Additionally, it would be good to see how this hook interfaces with the abalone, since it is quite unclear. Could the authors provide a picture from a typical experiment? In lines 136-137, the authors state that the hook is 3D printed. Could the authors provide more details, e.g., what printer and material were used.

- Figure 8 doesn't provide necessary details of the setup. It is way too simplified. Where is the aquarium tank? Is the aquarium tank used in the experiments the same as the one used for housing (that was mentioned in section 2.2.1)? Could the authors also provide a picture of the setup with an abalone during a typical test?

- The method for quantifying the abdominal foot area seems a bit suspicious. Why didn't the authors just directly measure the area for each abalone used? There are plenty of techniques that could be used to digitally measure the area from pictures (like the one in Figure 1a). There were only 5 individuals used in the study, so this would be very feasible. There are also many techniques for measuring the area simultaneously with the force.

- Table 1 only shows 2 repetitions of the force measurements for each abalone and each substrate. However, line 158 states that the measurements were carried out 5 times.

- Is Table 2 only for one abalone? The authors should provide ALL of the data obtained and used in the figures, either in the tables in the text or as supplemental information.

- What statistical test was used to produce the results of Table 3? Also, in lines 199-202, the authors state that the differences for two substrates are significant when compared to the smooth substrate. Can the authors state what they define as significant? Is it p-value of 0.05? The exact statistics need to be explained. How were the measurements averaged (within each individual or across all individuals)?

- Figure 10 shows the authors' hypothesis as to why the adhesive stress on the block pattern glass plate is higher than in the smooth plate. However, the authors don't report any direct measurements of this. Is the foot really in contact with all of the features of the block pattern plate? If so, could the authors provide evidence, for example, an image using frustrated total internal reflection, or any other technique for visualizing contact. If this hypothesis about increased surface area is true, then you would also observe higher stresses for the fine and coarse frosted glass plates since these also have increased surface area. I suggest the authors calculate the increased surface area of all the different substrates and use this to estimate the true contact areas of the abalones and their abdominal feet.

- Figure 12 shows the authors' hypothesis as to why the adhesive stress on the small lattice pit glass plate is higher than in the smooth plate. However, just like in Figure 10 and 11, the authors don't provide evidence for this, which can be easily achieved by imaging the contact area directly using, for example, frustrated total internal reflection. The height of the features on the block pattern glass plate (0.5mm) and the small lattice pit glass plate (0.8mm) are similar, so it is difficult to imagine why the abalone foot would adhere to one differently than the other.

- Lines 248-250 provide a conclusion about the morphology of the abdominal foot, but this is very simplistic. How many striped folds are found on the foot surface? What is the geometry of the folds? What are the heights of the folds? What is the material of the inner, middle, and outer layers and how do they compare to each other? I think a scanning electron microscope image would greatly benefit such an analysis. Or some scans of the surface features using other optical techniques, such as optical coherence microscopy or white light interferometry. I don't think the authors properly characterized the foot morphological properties.

- Lines 255-263 provide a conclusion regarding the ability for the abdominal foot to conform to the quadrangular conical glass plate. However, there is insufficient evidence provided. How flexible is the material of the inner layer of the foot? Is there also a mucus present in the contact region that could help with conformability? This is typically found in aquatic adhesive systems.

- Lines 264-266 provide a conclusion about how the foot cannot conform to micro-scale roughness. However, it is very unclear what the authors mean by the 'stretching change' of the abdominal foot and how this contributes to them not being to conform properly to micro-roughness.

Minor issues:

- Line 42, what is meant by 'adhesion organisms'? Is it meant to say 'organs' or 'organisms exhibiting adhesive capabilities'? I suggest the authors be more clear.

- Lines 45-46, this statement is quite vague and doesn't seem to add anything to the paragraph. Could the authors be more specific about how observations and experiments help reveal mechanisms? Isn't mathematical, physical, and/or chemical modelling also needed?

- Lines 47-48, what are 'typical adhesion organisms'? What makes an organism 'typical'?

- Lines 48-49, I don't think it is necessary to include the university or research institute for the citations in this paragraph.

- Lines 51-52, what is considered 'good adhesion'? Is there a certain force or stress above which the adhesion is considered 'good'? This is too vague and subjective of a statement (also lines 55-56 and 57-58). Also, 'throatfish' is not the name of the organism in references 16-17. Those papers studied northern clingfish.

- Lines 58-59, can the authors provide some perspective regarding how 'strong' the adhesive capacity of the abalone's abdominal foot is? Maybe compared to the abalone's weight? Or compared to other adhesive organisms or adhesive systems in every day life?

- Figure 1b should have a scale bar as well.

- Lines 156-157, it is unclear what is meant by 'test period'. Does the whole pulling experiment take place across 24 hours or is it that the abalones are given 24 hours to adhere to the glass plate and then tested?

- Line 184, Figure 9 is a box plot and not a histogram.

- For Figure 9, what do the ranges of the black lines represent? Is it standard deviation?

- Lines 199-202, the authors state that the differences for two substrates are significant when compared to the smooth substrate. Can the authors state what they define as significant? Is it p-value of 0.05?

- Line 205, what is meant by a rotation angle being 'gentle and softly'? This seems like a typo and also needs to be more specific.

Comments on the Quality of English Language

The writing needs to be greatly improved. There are a lot of vague statements that don't contribute anything useful to the manuscript. In the 'minor issues' feedback in my previous text, I point out some examples, but there are many more throughout the text.

Reviewer 2 Report

Comments and Suggestions for Authors

The paper illustrates adhesion forces than can be generated by a species of Abalone on various substrates. I think that the experimental work is well done and it can make a suitable contribution to the Journal Biomimetics. However, the presentation and the interpretations leave much to be desired as I detailed in my comments below. These are major issues and I highly recommend that a native speaker should go over the manuscript prior to resubmission.

The English must be improved and the formulation of “generalized biological concepts” avoided or specified. Many of these concepts are false, redundant or trivial. I will not detail all instances, I reviewed the first 2 pages (introduction) rather thoroughly, but I will not point out each and every erroneous sentence or linguistic issue as there are too many. This requires thorough proofing by the authors prior to (re-)submission and is not the duty of the reviewer.

Two examples of “concepts”:

Page 1, lines 34 and 35: “Continuous evolution”. In a Lamarckian sense (?), this is not true, there are sudden mutations in the genome that are believed to largely cause evolution.

Idem: “Adhesion is the basic ability of a variety of animals in nature” What is a “basic ability”??, the same is true for walking, flying, swimming etc…. “Animals in nature” is redundant, “animals” alone is good enough.

Other examples of poor language and vague formulations:

Page 2: Formulations such as “achieved a lot of results” (lien 48), “carried out a lot of research” and others are poor language and should not be part of a scientific publication.

Line 48_ “Ditsche et al.” not „Petra ditsche et al.“ (…) “found that the surface of the sucker” (line 49)- Please specify which sucker is meant in the first sentence. There is no such thing as “THE biological sucker”

Page2, various lines: “from the University of Washington”, “from National Tsinghua University in Taiwan”, “from the University of California”, “from China University of petroleum”, “from Shanxi Agricultural University”

It is very unusual to credit the institutions, the name of the first author or just the citation [number in bracket] is sufficient.

Line 76: “tensile tests” or “tensile testing”

Line 84, this instance and others. Please always write genus and species namens in italics, e.g. Haliotis discus hannai

Line 90: How can the foot of one single abalone individual be 1915~2760 mm2?? (…) on further reading the msausrcipt I finally saw (on page 6) that six abalones were used. This information must be provided much earlier.

Lines 90-91 and 109-112: especially clumsy sentences, please rephrase. At this point I stop pointing out these issues though there are many to come, e.g. lines 204 and 205 as well as before and after that.

Lines 112-114: the authors surely also chose a glass plate for the experiment as this allows for observing the foot’s area (changes) during tensile tests. Please mention this.

Page 4 and 5: Figures 2-5: these four figures can well be condensed to one figure by rearranging and combining Fig.5 with the details of figures 2 and 3 and omitting the rest of figure 2and 3 (redundant) and the whole figure 4 (also redundant).

Lines 139-140: very poor language and way too much information in the figure caption which needs to be spelled out in the main text.

Figures 6 and 7 should be combined.

Line 147: Please specify the manufacturer or model of the testing machine

In the whole Material and Method section, I still do not understand what force is measured. The maximal force generated by the abalones prior to being pulled off? This vital information is lacking.

Figure 8: This sketch is somewhat erratic. (1) What is a “lifter” “cross-heads”? (2) I assume that the red line are strings that connect the force sensor to the hooks that hold the shell. If so, which strings were used? Were these stiff enough in comparison to tensile forces generated by the abalones?

Results pages 7 and 8: I do not understand the values for mass/g of the ablones. Are there 6 abalones used for ever repetition. So in all 30 individuals?? I suggest that the tables 1 and 2 are moved into a supplement for further reference as these are very hard to grasp in the main text. The statistical significance information in table 3 should be combined with figure 9 by indicating statistical differences or non-differences with lower case letters as explained in e.g.

https://www.researchgate.net/figure/Different-letters-indicate-statistically-significant-differences-P005_fig2_259395968.

This can be easily done here and will largely facilitate the discussion.

Results (overall): Why are there no figures with photographs that illustrate the foot of the abalone on various glass substrates? This could well support the interpretation in figures 10-12, which are, otherwise, purely speculative.

Lines 224-225 and figure 12: what is an independent sucker system? Does this imply that underpressure is formed when detaching the foot of the abalone? This cannot be proven by results shown here. Are there indications that the adhesion area is increased by this pattern? This would also explain the higher pull-off force.

Comments on the Quality of English Language

The English must be significantlx improved. There are many instances of clumsy, or vague sentences as detailed in the rest of the reviewer report.

Reviewer 3 Report

Comments and Suggestions for Authors

Here the authors explore abalone adhesion beyond smooth surfaces, observing the stripe-shaped folds on the abdominal foot. Abalone exhibits similar adhesion on smooth, fine frosted, coarse frosted, and quadrangular conical glass plates. However, significant variations are observed on small lattice pit and block pattern glass plates, where the abdominal foot's flexibility enables effective adhesion to block patterns and the formation of independent sucker systems in each small lattice pit, respectively. The manuscript is interesting. It can be accepted after following corrections.

Some comments

1.     While the paper conducts a comprehensive analysis of the adhesion qualities of abalone abdominal foot on various force measurement plates, there is no specific research topic or hypothesis at the start of the study. It would improve the writing by fully stating the research aim, resulting in readers understanding the study's objectives.

2.     A more in-depth explanation of the results' practical implications would improve the paper. How can knowledge about abalone adhesion processes on various surfaces be utilized in real-world situations, like as material design or industrial applications?

3.     Please combine figures 2, 3, and 4 to create a single figure. The same proposal is valid to Figures 10, 11, and 12.

4.     The scale bar is missing for figure 1b.

5.     Please suggest a more realistic and representational scheme in Figure 8.

6.     Explaining the data in detail and providing a process for various surfaces is necessary.

7.     Recognizing any challenges or possible sources of mistake would improve transparency in the study.

8.     The use of graphical abstracts improve the research's accessibility by providing readers with a concise summary of the essential results.

9.     Readers without a specialist background may struggle to understand the technical concepts and terminology linked to adhesion processes. Consider offering better explanations or definitions to help a larger audience understand.

10.  The research focuses on the effect of various surface morphologies on abalone adhesion, but it does not provide a thorough evaluation of other elements that may affect adhesion in more complicated, real-world circumstances. The research focuses on adhesion on dry surfaces and does not account for the possible impact of water or other environmental conditions. Abalones often function in aquatic habitats, and their adhesion processes may vary depending on the surroundings.

Comments on the Quality of English Language

 Minor editing of English language required

Round 2

Reviewer 1 Report

Comments and Suggestions for Authors

I thank the authors for addressing my concerns. However, I still believe that their interpretations are not strongly supported by their data. Therefore, the language around the description of the elasticity of the foot and its ability to conform to structures should emphasize that it is just a hypothesis and would need further experiments to confirm.

Comments on the Quality of English Language

The authors addressed the minor issues I had pointed out during the first draft.

Reviewer 3 Report

Comments and Suggestions for Authors

 Accept in present form

Author Response

The reviewer has no comments and suggestions for authors.